# Enhanced Soil Fertility and Carbon Sequestration in Urban Green Spaces through the Application of Fe-Modified Biochar Combined with Plant Growth-Promoting Bacteria

**DOI:** 10.3390/biology13080611

**Published:** 2024-08-12

**Authors:** Guoyao Niu, Chiquan He, Shaohua Mao, Zongze Chen, Yangyang Ma, Yi Zhu

**Affiliations:** 1School of Environmental and Chemical Engineering, Shanghai University, Shanghai 200444, China; guoyaoniu@163.com (G.N.); shmao1234@163.com (S.M.); czzsuccess@163.com (Z.C.); 16609453783@163.com (Y.M.); 2Key Laboratory of National Forestry and Grassland Administration on Ecological Landscaping of Challenging Urban Sites, Shanghai Engineering Research Center of Landscaping on Challenging Urban Sites, Shanghai Academy of Landscape Architecture Science and Planning, Shanghai 200232, China

**Keywords:** carbon sequestration, enzyme activity, Fe-modified biochar, soil fertility, soil restoration

## Abstract

**Simple Summary:**

This study investigated the effects of plant growth promoting bacteria (*Bacillus clausii*) and Fe-modified biochar on soil fertility increases and mechanisms of carbon sequestration. Additionally, the impact on C-cycling-related enzyme activity and the bacterial community was also explored. The study results demonstrate that in comparison to the individual application of FeB and BC, the FeBBC treatment significantly relieves soil alkalization and enhances soil alkali-hydro nitrogen content and aggregate stability (particle size > 0.25 mm), thereby contributing to improved soil fertility and ecological function. Additionally, all biochar treatments exhibit higher soil organic carbon, thereby increasing organic carbon sequestration, particularly in the FeBBC treatment. Compared to a single ecological restoration method, FeBBC treatment can improve soil fertility and carbon sequestration, providing important reference values for urban green space soil ecological restoration projects.

**Abstract:**

The soil of urban green spaces is severely degraded due to human activities during urbanization, and it is crucial to investigate effective measures that can restore the ecological functions of the soil. This study investigated the effects of plant growth promoting bacteria (*Bacillus clausii*) and Fe-modified biochar on soil fertility increases and mechanisms of carbon sequestration. Additionally, the effects on C-cycling-related enzyme activity and the bacterial community were also explored. Six treatments included no biochar or *Bacillus clausii* suspension added (CK), only *Bacillus clausii* suspension (BC), only biochar (B), only Fe-modified biochar (FeB), biochar combined with *Bacillus clausii* (BBC), and Fe-modified biochar combined with *Bacillus clausii* (FeBBC). Compared with other treatments, the FeBBC treatment significantly decreased soil pH, alleviated soil alkalization, and increased the alkali-hydro nitrogen content in the soil. Compared to the individual application of FeB and BC, the FeBBC treatment significantly improved aggregates’ stability and positively improved soil fertility and ecological function. Additionally, compared to the individual application of FeB and BC, the soil organic carbon (SOC), particulate organic carbon (POC), and soil inorganic carbon (SIC) contents for the FeBBC-treated soil increased by 28.46~113.52%, 66.99~434.72%, and 7.34~10.04%, respectively. In the FeBBC treatment, FeB can improve soil physicochemical properties and provide bacterial attachment sites, increase the abundance and diversity of bacterial communities, and promote the uniform distribution of carbon-related bacteria in the soil. Compared to a single ecological restoration method, FeBBC treatment can improve soil fertility and carbon sequestration, providing important reference values for urban green space soil ecological restoration.

## 1. Introduction

Urban green spaces refer to land areas in urban environments where natural and artificial vegetation are the predominant forms of existence [1]. The soil in urban green spaces is a vital constituent of terrestrial ecosystems, playing a crucial role in mitigating climate change and sequestering carbon in the soil [2]. The soil in urban green spaces can increase its nutrient content and enhance its carbon storage capacity through atmospheric carbon dioxide precipitation or decomposition of plant waste [3]. Nevertheless, during the rapid urbanization process, the soil of these urban green spaces is commonly affected by multiple direct or indirect human disturbances or obstacles, resulting in problems such as compaction, high pH, and low soil nutrient and organic matter content, profoundly impacting its ability to fulfill ecological functions [4,5]. Hence, investigating measures to alleviate the degradation of soil fertility and enhance soil carbon sequestration in urban green spaces has emerged as a crucial research focus.

Biochar is a solid carbonaceous material produced by converting carbon rich biomass through thermochemical methods under anaerobic or hypoxic conditions. Previous studies have demonstrated that adding biochar to soil effectively improves soil fertility and increases carbon sequestration [6,7]. Recent research has focused on the functional modification of biochar to enhance its potential applications in carbon sequestration, soil fertility improvement, and environmental pollution remediation [8,9]. Wen et al. [10] indicated that, compared with biochar, iron-modified biochar significantly reduces the pH value and has more potential in alleviating soil alkalization. Wu et al. [11] found that Fe (III)-biochar has a larger specific surface area, pores, and more functional groups, which helps enhance its ability to restore soil ecological functions. Furthermore, Zhang et al. [12] found that the application of Fe_3_O_4_-modified biochar in soil resulted in higher organic carbon content and carbon-nitrogen ratio compared to the addition of raw biochar and Fe_3_O_4_ alone. Based on these findings, taking advantage of Fe-modified biochar for soil recovery in urban green spaces has great potential [9,13].

Plant Growth-Promoting Bacteria (PGPB) can promote plant growth through direct and indirect effects, thereby increasing soil carbon sequestration [14]. PGPB can directly promote plant growth through the synthesis of growth hormones (IAA) or through the effects of phosphorus solubilization, nitrogen fixation, and potassium solubilization [15]. In addition, PGPB can indirectly promote plant growth by improving soil structure and inhibiting the growth of pathogens [16]. *Bacillus* sp. is the bacterium that is most frequently utilized for promoting plant growth. Some studies have demonstrated that the inoculation of *Bacillus* sp. can effectively increase soil nutrient and carbon sequestration [17]. *Bacillus clausii* is a microorganism that has been recognized for its ability to promote plant growth. Li et al. [18] confirmed the production of extracellular polysaccharides by *Bacillus clausii* isolated from saline-alkali soil in Xinjiang. The *Bacillus clausii* B8 strain isolated from soil by Oulebsir-Mohandkaci et al. [19] can produce siderophore, HCN, and IAA, leading to a significant improvement in rapeseed seed germination rate. Although *Bacillus clausii* has been recognized for its plant growth-promoting attributes, its potential role as a beneficial bacterium in improving soil fertility and carbon sequestration within the context of urban green spaces is yet to be comprehensively elucidated.

Ecological remediation of soils by plants has been widely used and is often attributed to changes in plant and soil microbial communities [20]. However, the comparison of the effects between biotic and abiotic soil ecological remediation materials has not been fully studied. Previous research has demonstrated that biochar can serve as a shelter for bacteria by providing a favorable pore structure and specific surface area, while also supplying essential nutrients for bacterial growth [21,22]. Furthermore, biochar has been observed to enhance the physicochemical properties of soil, stimulate microbial activity, and thus positively affect the soil carbon pool [23]. For example, Jabborova et al. [24] indicated that co-inoculation of biochar and rhizobacteria significantly increased soil SOC and nutrient content. Ren et al. [25] indicated that the combined application of PGPR and biochar significantly increased the soil bacterial diversity index. However, most research mainly focused on the combined biotechnology of biochar and PGPB. In contrast, the effect of Fe-modified biochar combined with *Bacillus clausii* on improving soil fertility in urban green spaces and carbon sequestration is still unclear.

To fill this knowledge gap, this study aimed to investigate the effects of applied *Bacillus clausii*, biochar, Fe-modified biochar, biochar combined with *Bacillus clausii*, and Fe-modified biochar combined with *Bacillus clausii* on soil fertility in urban green spaces and carbon sequestration and explored the changes in soil C-cycling-related enzyme activity and the bacterial community. We hypothesized that (1) compared to their individual application, the combination of Fe-modified biochar and *Bacillus clausii* will result in improving soil fertility; (2) the combination of Fe-modified biochar and *Bacillus clausii* could significantly enhance the ability of soil carbon sequestration; and (3) the combination of Fe-modified biochar and *Bacillus clausii* can improve enzyme activity and bacterial community composition related to carbon cycling in the soil.

## 2. Materials and Methods

### 2.1. Preparation of Soil Samples, Biochar, Bacillus clausii, and Ryegrass Seeds

The soil samples were collected from the green spaces of Shanghai University (31°19′34″ N; 121°24′22″ E). Using a multipoint mixing method, four sampling points were used to collect soil samples from the surface layers (upper 20 cm). The drying method was used to measure the soil moisture content of the field soil samples immediately upon their return to the laboratory. Before being used for the incubation experiment, ensure that the soil is thoroughly mixed and dried using air conditions and passed through a 2 mm sieve. The physicochemical properties of the soil are shown in Appendix A.

The biochar (B) was provided by Shanghai Carbon Suo Era Environmental Technology Co., Ltd., Shanghai, China. The biochar was derived from apple wood and obtained using a continuous operation reactor at a temperature of 400 °C for 5 h. The preparation of Fe-modified biochar (FeB) is based on the modification method mentioned by Liu et al. [26]. All the biochar samples were passed through a 2 mm sieve before being used. *Bacillus clausii* CICC 21104 was provided by the China Center of Industrial Culture Collection (http://www.china-cicc.org/cicc/detail2/?sid=3305, accessed on 14 January 2023), Beijing, China. The ryegrass seeds (*Lolium perenne* L.) were provided by Beijing Hejia Eco-technology Co., Ltd., Beijing, China.

### 2.2. Bacterial Inoculum Preparation

Nutrient broth (CM0002, China Center of Industrial Culture Collection, Beijing, China) culture medium was the medium used for culture of *Bacillus clausii*. *Bacillus clausii* was inoculated into sterile medium to cultivate and expand at 37 °C for 48 h. We regulated the suspension of *Bacillus clausii* to 1 × 10^8^ CFU mL^−1^ and then used it as a standard inoculum.

### 2.3. Experimental Design

On 1 July 2023, an indoor potted plant experiment was conducted at the artificial climate laboratory of the School of Environmental and Chemical Engineering, Shanghai University. The six treatments are shown in Table 1. Each treatment had three replicates of pots. Firstly, we filled 1200 g soil into pots with an inside diameter of 12 cm and a height of 22 cm. For the CK treatment, no biochar or *Bacillus clausii* suspension was added. For the *Bacillus clausii* (BC) treatment, we diluted 10 mL of *Bacillus clausii* suspension with sterile water to 100 mL, and then fully mixed with the soil in the pot. For the biochar (B) or Fe-modified biochar (FeB) treatment, we fully mixed the biochar or Fe-modified biochar with the soil in the pot at a rate of 2% (*w*/*w*), respectively. For the biochar combined with *Bacillus clausii* (BBC) or Fe-modified biochar combined with *Bacillus clausii* (FeBBC) treatment, we fully mixed the same above-mentioned doses of biochar or Fe-modified biochar and *Bacillus clausii* suspension with the soil in the pot. To ensure consistency, add the same doses of sterile water for treatments without *Bacillus clausii* inoculation.

The ryegrass seeds (*Lolium perenne* L.) underwent sterilization using a 10% H_2_O_2_ solution for 20 min. After sterilization, the seeds were thoroughly rinsed with deionized water until clean and soaked in deionized water for 2 h prior to sowing. All pots were watered with deionized water every two days, and the soil moisture content was maintained at about 60% field soil moisture content. After ten days of growth, 50 seedlings of the same growth were retained in each pot. The incubation experiment lasted 56 days at a controlled temperature of 25 ± 2 °C.

### 2.4. Samples Analysis

#### 2.4.1. Characterization of *Bacillus clausii* and Biochar Samples

The pH, element contents (C, N, and H), surface area, pore volume, and pore diameter were selected to characterize the basic properties of biochar samples. A scanning electron microscope with energy dispersive spectrometer (SEM-EDS), Fourier transform infrared (FTIR) spectra, X-ray diffraction (XRD), and X-ray Photoelectron Spectroscopy (XPS) were used to analyze the structural features of biochar samples.

The growth promoting characteristics of *Bacillus clausii* were evaluated, including IAA production, ammonia production, phosphate solubilization, nitrogen fixation, siderophore production, and the production of carbonic anhydrase. Inoculate *Bacillus clausii* into the liquid B4 medium for biomineralization experiments to evaluate the effect of *Bacillus clausii* on carbonate precipitation. The FTIR spectra, XRD, and SEM-EDS were used to analyze the surface morphological characteristics and the composition of the mineralization products.

#### 2.4.2. Analysis of Plant and Soil Samples

Soil and plant samples were taken after the incubation experiment to evaluate the effect of soil treatments on plant biomass and soil. A portion of the freshly collected soil was stored in a refrigerator at −20 °C for the analysis of C-cycling enzyme activity. To analyze the diversity and abundance of bacterial communities, a portion of fresh soil was gathered in sterilized bags and kept it in a refrigerator at −80 °C. The remaining fresh soil was dried under air conditions and then analyzed for soil properties after being sieved.

The basic soil properties pH, alkali-hydro nitrogen, available phosphorus, available potassium, available Fe, free Fe oxide (Fed), and soil aggregates were measured to evaluate the effects of soil treatment on soil fertility and structure. The contents of soil organic carbon (SOC), particle organic carbon (POC), KMnO_4_-oxidized organic carbon (EOC), dissolved organic carbon (DOC), and soil inorganic carbon (SIC) were measured to evaluate the effects of soil treatment on soil carbon sequestration. This study selected invertase and β-glucosidase to assess the impact of soil treatment on C-cycling enzyme activity. The Appendix A provides detailed information on the measured methods.

#### 2.4.3. Analytic Method for Bacterial Community Analysis

Soil samples were processed using the MagAttract^®^ PowerSoil^®^ Pro DNA Kit (QIAGEN, Hilden, Germany) following the manufacturer’s instructions. To analyze bacterial diversity, the V3-V4 hypervariable region of the 16S rRNA genes were amplified with universal primers 338F (5′-ACTCCTACGGGAGGCAGCAG-3′) and 806R (5′-GGACTACHVGGGTWTCTAAT-3′). The raw sequencing data have been deposited into the NCBI Sequence Read Archive (SRA) database (Accession Number: PRJNA1069871).

### 2.5. Statistical Analysis

To assess significant differences in plant biomass, soil physicochemical properties, soil nutrient content, soil carbon component content, and enzyme activity were examined using a one-way analysis of variance (ANOVA) with Duncan’s multiple range test (*p* < 0.05). The results were presented as the mean ± standard error of the mean of three replicates. The Pearson correlation analysis method was employed to explore the relationship between soil and environmental factors. A co-occurrence network was constructed using Gephi to visualize the associations between the diversity of different bacterial orders and ecological aspects. A correlation between two nodes was considered to be statistically robust if the Spearman’s correlation coefficient was over 0.8 or less than −0.8 and the *p*-value was less than 0.05. The software of Excel 2016 and Origin 2022 was used to produce the tables and figures.

## 3. Results

### 3.1. Changes in Characteristics of Fe-Modified Biochar

The physicochemical properties of biochar are listed in Appendix A. The results showed that the pH and elemental content (C, N, and H) of FeB were significantly decreased. In addition, the specific surface area and Fe content of FeB increased to 9.35 m^2^ g^−1^ and 22.63 g kg^−1^, respectively. SEM analysis clearly revealed the porous structure of biochar, and after Fe-modification, nanoparticles were observed to be loaded into the pores of FeB (Appendix A). FTIR spectroscopy revealed the presence of new absorption peaks at 650 cm^−1^ and 460 cm^−1^ in FeB (Appendix A). XRD analysis demonstrated the presence of characteristic peaks of Fe_2_O_3_, Fe_3_O_4_, and FeO on the surface of FeB (Appendix A). After modification, the C-O peak of FeB disappeared, while the peaks corresponding to C-C, C=O, and O-C=O increased (Appendix A).

### 3.2. Plant Growth-Promoting Characteristics and Biomineralization of Bacillus clausii

The results showed that *Bacillus clausii* has various plant growth promoting characteristics, such as indole-3-acetic acid (IAA) production, phosphorus solubilization, nitrogen fixation, siderophores production, and carbonic anhydrase production (Appendix A). In the biomineralization experiment (Appendix A), the SEM-EDS analysis of the mineralized product (MP) revealed that the main elements of the MPs were C, O, and Ca (Appendix A). The absorption peaks at 875 cm^−1^, 1401 cm^−1^, and 1436 cm^−1^ were the absorption peaks of hexagonal calcium carbonate and calcite, respectively (Appendix A) [27]. XRD spectrum showed (Appendix A) that the diffraction peak at 2θ = 29.4° corresponds to the (104) crystal plane of calcite [28].

### 3.3. Changes in Soil Physicochemical Properties and Fertility of FeB Combined with BC

The effects of various treatments on the physicochemical properties of the soils differ significantly (Table 2). The soil pH significantly decreased in the BC, FeB, and FeBBC treatments. However, the soil pH significantly increased in the B and BBC treatments. Compared with other treatments, in the FeBBC treatments, both the aboveground weight of *Lolium perenne* L. and the alkali-hydro nitrogen content in the soil remained consistently high (Appendix A). However, FeBBC treatment had a negative effect on both available P and K compared to the BBC treatment. Furthermore, the FeB treatment exhibited a significant decrease in soil available Fe content when compared to the CK. The available Fe and Fed content in the FeBBC-treated soil is significantly increased compared to the BBC treatment, indicating that the combination of FeB and BC positively affects the available Fe and Fed content. After 56 days of incubation, compared with other treatments, there was a significant increase in the specific gravity of soil aggregates (particle size > 0.25 mm) that was observed for the FeBBC treatment (Figure 1f).

### 3.4. Changes in Soil Carbon Components for FeB Combined with BC

The combination of FeB and BC resulted in elevated levels of SOC content compared to other treatments (Figure 1a). Compared with BC or unmodified biochar (B and BBC) treatments, the higher content of POC was observed for Fe-modified biochar treatments (FeB and FeBBC) (Figure 1c). Furthermore, including BC, FeB, BBC, and FeBBC treatments resulted in significantly higher levels of DOC (Figure 1b). Regarding EOC, the Fe-modified biochar treatments (FeB and FeBBC) exhibited a slight decrease in EOC content compared to the unmodified biochar treatments (B and BBC). However, this difference was not statistically significant (Figure 1d). Among all the treatments, BC and FeB treatments showed lower SIC content, whereas BBC treatment exhibited higher SIC content. Furthermore, when comparing FeB with FeBBC treatment, a significant increase in SIC content was observed in the latter (Figure 1e).

### 3.5. Changes in Soil C-Cycling-Related Enzyme Activity and Soil Bacterial Diversity for FeB Combined with BC

Among all the treatment groups, the FeB and FeBBC treatments exhibited lower levels of β-glucosidase (Figure 2a). Similar trends were observed in terms of invertase activity among all treatments. The B and FeBBC treatments displayed higher levels of invertase activity, while no significant differences were observed between the other treatment groups (Figure 2b).

The changes in the alpha diversity of soil bacterial communities are shown in Table 3. The results of this study suggested that the gene sequence utilized accurately reflects the composition of bacterial communities in the soil samples. This conclusion is supported by the high Goods coverage, ranging from 99.59% to 100.00%. The FeBBC treatment resulted in a diverse bacterial community, as demonstrated in Table 3. However, the BBC treatment provided the least diversity in the community. Likewise, the number of ASVs correlates with the trend of similarity in the alpha diversity index. The Venn diagram analysis revealed variations in the ASVs among different treatments (Appendix A). The number of treatment-specific ASVs ranged from 1684 to 2292, and the highest value was observed for the FeBBC treatment, which suggests that the FeBBC treatment improved the environment for microbial survival. Furthermore, a total of 533 ASVs were found to be shared among all treatments, as depicted in Appendix A. These core bacterial groups primarily included *Proteobacteria*, *Acidobacteriota*, *Actinobacteruota*, *Chloroflexi*, and *Firmicutes*.

### 3.6. Changes in Bacterial Community Composition for FeB Combined with BC

The relative abundance of dominant phyla in bacterial communities and the heatmaps of 30 dominant bacterial orders are shown in Figure 3, respectively. The focus of this study will be on taxonomic groups that are closely linked to carbon decomposition/fixation in soil. The result showed that the dominant C-cycling-related bacterial taxa included the Proteobacteria, Acidobacteriota, Actinobacteriota, Chloroflexi, Firmicutes, Myxococcota, and Gemmatimonadota. Compared with unmodified biochar, the relative abundance of Proteobacteria, Firmicutes, and Gemmatimonadota increased with Fe-modified biochar treatment. Actinobacteriota and Chloroflexi were less abundant in the FeBBC-treated soil than in the BBC-treated soil. Moreover, the abundance of Acidobacteriota for the unmodified biochar treatment was higher than that for the Fe-modified biochar treatment. The distribution of C-cycling-related bacteria treated with FeBBC was uniform compared to adding FeB or BC alone. Notably, the FeBBC treatment exhibited a higher relative abundance of *Rhizobiales*, *Burkholderiales*, *Sphingomonadales*, and *Gemmatimonadales* than the BBC treatment. Additionally, the relative abundance of *Myxococcales* in the FeBBC-treated soil decreased in comparison to the FeB treatment (Figure 3b).

### 3.7. Exploring the Correlations between Environmental Factors and Bacterial Communities

Pearson correlation analysis (Figure 4a) revealed significant positive correlations between soil pH and available P, available K, and β-glucosidase activity. Moreover, SOC demonstrated significant positive correlations with the aboveground weight of *Lolium perenne* L., particulate organic carbon, and invertase activity. As illustrated in Figure 4b, the co-occurring network analysis revealed significant negative correlations between soil pH and *Chloroflexales* and *Sphingomonadales*. A significant negative correlation between SOC and *Myxococcales* was also revealed.

## 4. Discussion

### 4.1. Characteristics of Biochar and Fe-Modified Biochar

The main reason for the significant decrease in the pH value of FeB was the hydrolysis of Fe^3+^ on the surface of FeB [10]. This indicated that adding acidic FeB to soil will positively impact the pH, alleviating soil alkalization. In addition, an increase in the specific surface area of FeB and the porous structure observed via SEM indicates that FeB had more adsorption sites, which was beneficial for the survival of bacteria. FTIR spectroscopy revealed the presence of new absorption peaks at 650 cm^−1^ and 460 cm^−1^, indicating the successful loading of Fe_2_O_3_ and Fe_3_O_4_, which is also confirmed in the XRD analysis [29]. XPS spectroscopy shows that the changes in different functional groups on the surface of FeB indicate that the increased aromaticity of FeB and alterations in its surface characteristics can impact its interaction with various soil environmental factors, consequently influencing the composition of bacterial communities within the soil.

### 4.2. Characteristics of Bacillus clausii

The results in Appendix A indicate that *Bacillus clausii* exhibited positive traits for plant growth promotion. The mineralization experiment’s products were mainly carbonates, as determined with SEM-EDS, FTIR, and XRD characterization analysis. The findings of this study provide evidence that *Bacillus clausii* is capable of producing carbonic anhydrase, an enzyme that facilitates the conversion of CO_2_ into carbonate. This enzymatic activity is advantageous for the sequestration of inorganic carbon in soil, contributing to its overall carbon storage capacity.

### 4.3. Effect of FeB Combined with BC Improves Soil Physicochemical Properties and Fertility

In this study, the high pH of biochar itself and the hydrolysis of soluble alkaline substances led to an increase in soil pH in the B and BBC treatments, exacerbating soil alkalization. In accordance with our first hypothesis, both FeB and BC treatments had a positive effect on reducing the soil pH value. Moreover, compared with an individual application, the combined application of FeB and BC resulted in a more significant decrease in soil pH value. The hydrolysis of Fe^3+^ on the surface of Fe-modified biochar released a large amount of H^+^, and consequently reduced soil pH [10]. In addition, this phenomenon may also be attributed to FeB providing a favorable habitat for *Bacillus clausii*, which in turn secretes organic acids. These organic acids not only contribute to the reduction in soil pH but also facilitate the dissolution of inorganic phosphates that are present [30].

In this study, the aboveground fresh and dry weight of *Lolium perenne* L. were significantly increased in the FeBBC treatment (Appendix A). This is because the addition of Fe-modified biochar provided a larger contact area for plant roots to absorb nutrients from the soil, and improved plant growth under the action of plant growth promoting bacteria. Compared with BBC or FeB treatments, the highest alkali-hydro nitrogen content in soil was observed with the FeBBC treatment. This may be attributed to the ability of FeB to regulate the C/N ratio in the soil by altering the soil pH, thereby affecting the availability of nitrogen in the soil. Additionally, as shown in Appendix A, *Bacillus clausii* in the FeBBC treatment exhibited characteristics such as IAA production and nitrogen fixation, promoting plant growth and enhancing the effectiveness of nitrogen in the soil [31]. Comparatively, the utilization of FeBBC led to a significant decrease in the availability of soil P and K in comparison to the BBC treatment. These findings align with previous studies that observed reduced soil phosphorus availability upon adding Fe-modified biochar [11]. The decrease in the soil availability of P may have been related to the combination of phosphorus anions and iron hydroxides, which reduces the bioavailability of phosphorus in the soil. In addition, the low pH value of the soil and the adsorption of Fe-modified biochar may also be the reasons for the decrease in soil nutrients (P and K). The application of iron-modified biochar may inhibit the activity of iron reducing bacteria [32]. In addition, the adsorption of biochar and the binding of available Fe and phosphorus reduced the content of available Fe in the FeB treatment soil [33]. Compared with the BBC treatment, adding FeB and BC also led to a significant increase in available Fe content in the soil, which was attributed to the role of the siderophore produced by *Bacillus clausii* (Appendix A).

After 56 days of incubation, compared to the BBC treatment, the FeBBC treatment exhibited a significant increase in the proportion of aggregates (particle size > 0.25 mm). The result showed a negative correlation between pH value and Fed (Figure 4a), indicating that soil pH influenced the formation of Fed. The introduction of exogenous Fe oxides through Fe-modified biochar played a crucial role in forming macroaggregates in the soil. It possesses the ability to bind with soil organic carbon, resulting in the formation of organic iron composite colloids [34,35]. In addition, our research results indicated that the higher aboveground biomass of plants and the richness and diversity of the soil bacterial communities in the FeBBC treatment increased the production of bacterial and plant viscous substances, consequently promoting the formation of aggregates. Therefore, the results of this study indicated that the combination of FeB and BC had great potential in alleviating soil alkalization, increasing soil alkali-hydro nitrogen content, and improving soil aggregate structure.

### 4.4. Effect of FeB Combined with BC Improves Soil Carbon Sequestration

It is well-established that adding biochar to the soil has been proven to significantly increase SOC content by providing exogenous organic carbon [36]. The results of this study were consistent with the previous studies, and our findings demonstrated higher SOC content in all biochar treatments. In accordance with our second hypothesis, compared with individual application, the combination of FeB and BC significantly enhanced the content of SOC in soil, and SOC reached its highest values for the FeBBC treatment. The highest content of SOC in the FeBBC treatment may be attributed to the high specific surface area of FeB and the creation of a suitable environment for microbial survival by providing nutrients, increasing the colonization of bacterial communities, and promoting plant growth, thus increasing SOC content. In our study, the result showed a notable positive correlation between SOC and the aboveground weight of *Lolium perenne* L. (Figure 4a), confirming that *Lolium perenne* L. growth contributes to the accumulation of SOC.

Compared with CK treatment, BC treatment showed no significant difference in POC. However, the higher POC content observed with FeBBC treatment can be attributed to the high surface activity of the Fe oxide on the surface of Fe-modified biochar, which can increase the carbon content. It also strengthens the bonding between soil clay particles and organic molecules and forms water-stable soil aggregates, preventing microorganisms’ rapid degradation of POC [26]. It is worth noting that higher DOC content was observed in Fe-modified biochar treatments (Figure 1b). This observation may be attributed to the biological pathway of Fe^3+^ reduction, which had been previously shown to increase DOC content [37]. After 56 days of BC treatment, the concentrations of DOC and EOC continued to increase. This may be attributed to the fact that BC could promote the secretion of a large amount of low-molecular weight organic compounds by ryegrass roots and enhanced bacterial activity, and thus accelerated the decomposition of soil organic matter. In terms of EOC, after modification, the active functional groups on the surface of FeB were removed (Appendix A), and the pore volume and pore diameter were decreased (Appendix A). This may be the reason for the decrease in EOC content in Fe-modified biochar treatments compared to unmodified biochar treatments.

The higher SIC content with the BBC treatment may be explained by its higher soil pH (Figure 1e). The significant decrease in SIC content with the BC and FeB treatments may be attributed to the growth of *Lolium perenne* L. Evidence has shown that plant roots absorb inorganic carbon from the soil during their growth process and convert it into organic carbon to meet plant growth needs [38]. The combination of FeB and BC resulted in a higher SIC content than individual applications. This may have occurred because the added *Bacillus clausii* inoculant produced carbonic anhydrase, which was beneficial for converting CO_2_, generated by atmospheric or biological respiration into carbonates. In addition, a short-term study found that carbonic anhydrase had the highest activity at pH 7.5, and the presence of Fe^3+^ enhanced the activity of carbonic anhydrase [39]. Therefore, the results of this study indicated that compared with individual applications, the combination of FeB and BC can significantly increase the content of SOC, POC, and SIC, promoting soil carbon sequestration.

### 4.5. Effect of FeB Combined with BC Improves C-Cycling-Related Enzyme Activity and Bacterial Community

The activity of enzymes related to carbon cycling in soil serves as a crucial indicator reflecting changes in soil organic matter (SOM). The decrease in β-glucosidase activity observed with the FeB and FeBBC treatments may be attributed to the co-localization of carbon and microorganisms on the surface of FeB in the soil, improving carbon utilization efficiency and reducing the production demand for β-glucosidase [40]. Compared with CK, there was no significant difference in the invertase activity between the BC treatment and FeB treatment, while the invertase activity with the FeBBC treatment significantly increased. The higher bacterial community richness may be responsible for the increase in invertase activity during FeBBC treatment. Furthermore, it has been established that a soil environment with a neutral to slightly alkaline pH promotes increased bacterial and enzymatic activity [41].

Bacterial communities play a crucial role in urban soil ecosystems, and changes in their structure can significantly impact the cycling of elements urban green spaces. In this study, it was observed that the richness and diversity of bacterial communities were significantly lower in all treatments except for the FeBBC treatment. This finding aligns with a study conducted by Zheng et al. [42], which also reported a reduction in bacterial richness with the addition of biochar. In addition, the competition between the inoculated *Bacillus clausii* and the local microbial community is weak and the colonization rate is limited [43]. The dramatically higher richness and diversity of bacterial communities in the FeBBC treatment, which may be attributed to the fact that positively charged Fe oxides on the surface of Fe-modified biochar can combine with negatively charged bacteria through electrostatic attraction, making them less prone to leaching in the soil and increasing the bacterial colonization rate.

Relative to the unmodified biochar treatment, the dominant C-cycling-related bacterial taxa increased under Fe-modified biochar treatment included Proteobacteria, Firmicutes, and Gemmatimonadota. The relative abundance of *Proteobacteria* is the highest in soils that have high levels of labile organic carbon. The order *Rhizobiales* (Alphaprotobacteria) is genetically capable of degrading various plant organic compounds [44]. Compared with unmodified biochar, the higher relative abundance of *Rhizobiales* with FeBBC treatment indicated that the addition of FeBBC had a good promoting effect on plant growth. The order *Sphingomonadales* and *Burkholderiales* belong to the phylum Proteobacteria, which can utilize various organic compounds or more refractory compounds like aromatics in the soil as a carbon source [45,46]. Meanwhile, the result indicated that the abundance of *Sphingomonadales* negatively correlated with the soil pH (Figure 4b). These observations suggested that *Sphingomonadales* demonstrated higher activity in neutral and slightly alkaline environments. *Firmicutes* bacteria are particularly responsive to an increase in recalcitrant carbon sources in the soil [26]. Therefore, the input of stable organic carbon may promote the rapid growth of *Firmicutes* in FeBBC-treated soil. The colonization of *Bacillus clausii* was indicated by the relative abundance of *Bacillales* belonging to *Firmicutes* increasing during FeBBC treatment. *Gemmatimonadales* contain diverse genes associated with organic carbon metabolism; they can utilize carbon and play a significant role in soil carbon fixation [47]. Therefore, the increase in relative abundance of carbon cycling-related bacteria indicated that Fe-modified biochar treatment was more effective at improving soil environment (pH and fertility) than unmodified biochar treatment.

The Acidobacteriota is a crucial bacterial community in soil that plays a significant role in the degradation of complex organic compounds [48]. The Actinobacteriota includes various microorganisms such as *Micromonosporales* and *Gaiellales*. The order *Micromonosporales* could encode multiple enzymes that aid in the degrading of organic C compounds [44]. The order of *Gaiellales* had been proven to be involved in the degradation of polycyclic aromatic hydrocarbons [49]. They play an important role in the degradation of various labile and stubborn carbon compounds. The lower relative abundance of Acidobacteriota and Actinobacteriota with FeBBC treatment indicated potential carbon sequestration and reduction in carbon dioxide emissions. According to several studies, Chloroflexi were typically found in anaerobic or nutrient-deficient environments [50]. The degrading overall of complex organic compounds is associated with Chloroflexi’s relative abundance in agricultural soils [47]. In addition, the result indicated that the abundance of *Chloroflexales* had negatively correlated with the soil pH (Figure 4b), The lower relative abundance of Chloroflexi with FeBBC treatment indicated that the addition of FeB combined with BC could improve the soil environment and could enhance soil fertility and aeration. The results indicated a significant negative correlation between *Myxococcales* and SOC. It has been confirmed that the extracellular hydrolytic enzymes secreted by *Myxococcales* play a crucial role in promoting the decomposition of complex carbon sources [47]. The lower relative abundance of *Myxococcales* with FeBBC treatment indicated that it was beneficial for reducing soil carbon mineralization. Therefore, the FeBBC treatment has a higher rate of soil carbon sequestration because Acidobacteriota, Actinobacteriota, Chloroflexi, and the order *Myxococcales* have lower relative abundance than the BBC treatment.

## 5. Conclusions

In conclusion, this study innovatively combined Fe-modified biochar (FeB) with *Bacillus clausii* (BC) to restore the ecological function of urban green soil. The results showed that the combination of FeB and BC effectively alleviates soil alkalization, improves soil fertility, and increases soil carbon sequestration. Additionally, the combination of FeB and BC significantly increased the activity of invertase and the richness and diversity of bacterial communities in the soil, promoting the uniform distribution of carbon cycle-related bacteria in the soil. The results clearly indicate that the combined application of FeB and BC has greater potential in improving soil fertility and increasing soil carbon sequestration in urban green spaces compared to their individual applications. These results provide a reference solution for the future restoration of soil ecological functions and provide a solid theoretical basis for restoring the ecological functions of urban green space soil and achieving sustainable management of urban or agricultural soil.

## Figures and Tables

**Figure 1 biology-13-00611-f001:**
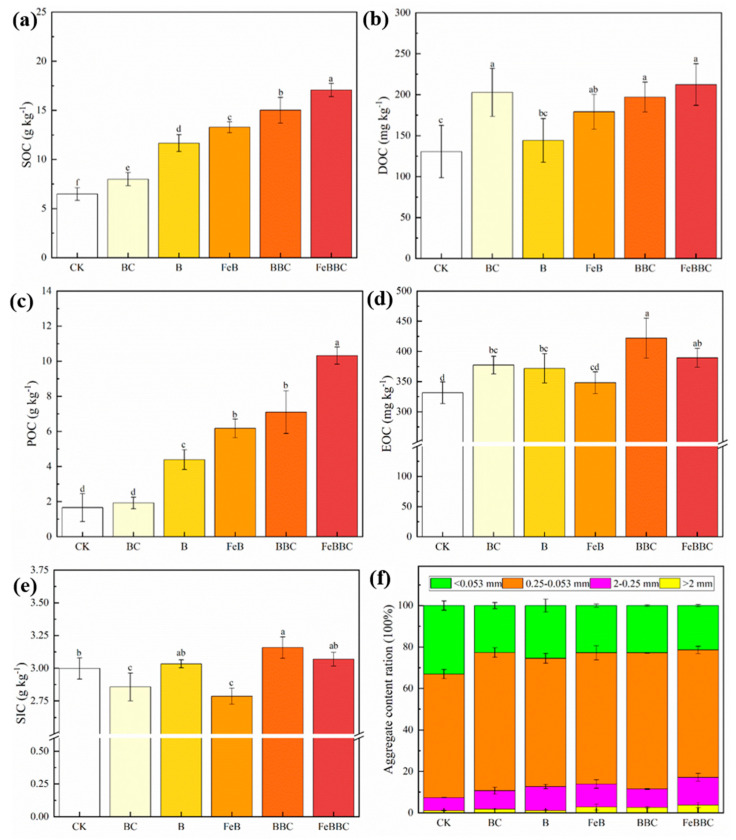
Mean contents of soil organic carbon (SOC) (**a**), dissolved organic carbon (DOC) (**b**), particulate organic carbon (POC) (**c**), KMnO_4_-oxidized organic carbon (EOC) (**d**), soil inorganic carbon (SIC) (**e**), and aggregate content ration (**f**) in the different treatments. Different letters indicate significant differences between the treatments and control (*p* < 0.05). CK: control; BC: *Bacillus clausii*; B: biochar; FeB: Fe-modified biochar; BBC: biochar combined with *Bacillus clausii*; FeBBC: Fe-modified biochar combined with *Bacillus clausii*.

**Figure 2 biology-13-00611-f002:**
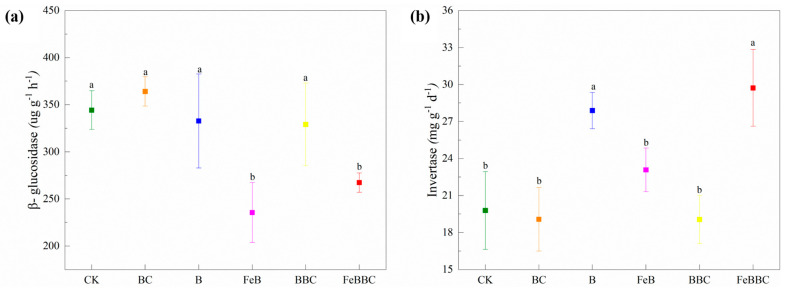
Changes in soil C-cycling-related enzyme activity in different treatments: β-glucosidase (**a**) and invertase (**b**). Different lowercase letters indicate significant differences between the treatment groups (*p* < 0.05). CK: control; BC: *Bacillus clausii*; B: biochar; FeB: Fe-modified biochar; BBC: biochar combined with *Bacillus clausii*; FeBBC: Fe-modified biochar combined with *Bacillus clausii*.

**Figure 3 biology-13-00611-f003:**
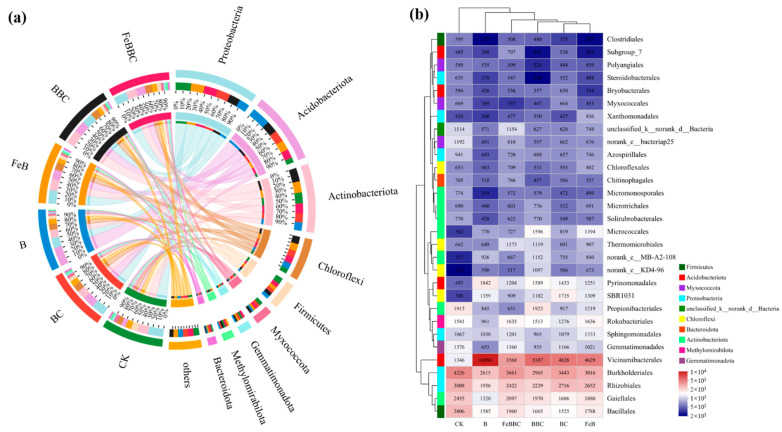
The relative abundance (**a**) of dominant species of bacterial communities. Heatmap diagram (**b**) of the dominant 30 bacterial orders in different treatments. CK: control; BC: *Bacillus clausii*; B: biochar; FeB: Fe-modified biochar; BBC: biochar combined with *Bacillus clausii*; FeBBC: Fe-modified biochar combined with *Bacillus clausii*.

**Figure 4 biology-13-00611-f004:**
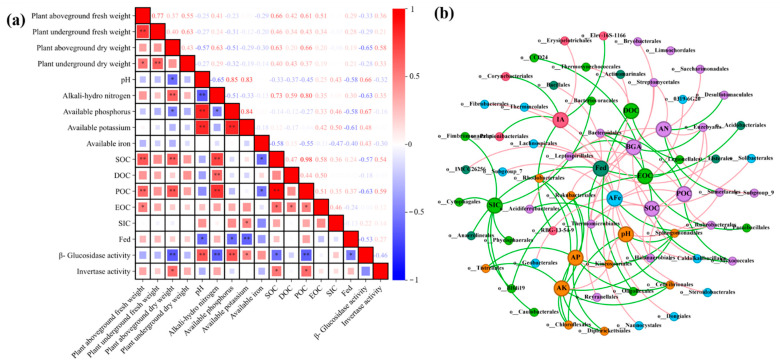
Pearson correlations (**a**) between soil environmental factors among all treatments. * and ** indicate significant correlations at *p* < 0.05 and 0.01, respectively. Co-occurring network (**b**) of the dominant 200 bacterial orders and environmental factors based on correlation analysis. The nodes in the network are colored based on modularity class. The connections stand for a strong (Spearman’s ρ > 0.8) and significant (*p* < 0.05) correlations. The red and green lines represent positive and negative correlations, respectively. Abbreviations: AP, available P; AN, alkali-hydro nitrogen; AK, available K; AFe, available Fe; IA, invertase activity; BGA, β-glucosidase activity.

**Table 1 biology-13-00611-t001:** Experimental treatments settings.

Samples	Biochar (%)	*Bacillus clausii* Suspension (1 × 10^8^ CFU mL^−1^) (mL)	*Lolium perenne* L.
CK	×	×	√
BC	×	10	√
B	2	×	√
FeB	2	×	√
BBC	2	10	√
FeBBC	2	10	√
Soil (g)	1200		
*Lolium perenne* L. (Number)	50		
Incubation cycle (d)	56		

Note: CK: control; BC: *Bacillus clausii*; B: biochar; FeB: Fe-modified biochar; BBC: biochar combined with *Bacillus clausii*; FeBBC: Fe-modified biochar combined with *Bacillus clausii*.

**Table 2 biology-13-00611-t002:** Soil physicochemical properties after incubation for 56 days.

Samples	pH	AP (mg kg^−1^)	AN (mg kg^−1^)	AK (mg kg^−1^)	AFe (mg kg^−1^)	Fed (mg kg^−1^)
CK	8.15 ± 0.02 ^b^	7.36 ± 0.82 ^c^	17.97 ± 1.07 ^d^	81.31 ± 2.75 ^c^	13.54 ± 0.50 ^b^	5.63 ± 0.13 ^ab^
BC	8.12 ± 0.02 ^c^	7.59 ± 0.23 ^bc^	20.77 ± 0.81 ^bc^	93.02 ± 2.53 ^b^	19.15 ± 1.35 ^a^	4.40 ± 0.68 ^c^
B	8.33 ± 0.01 ^a^	8.73 ± 0.60 ^a^	18.67 ± 2.65 ^cd^	124.34 ± 9.21 ^a^	12.64 ± 0.13b ^cd^	4.64 ± 0.55 ^c^
FeB	7.86 ± 0.02 ^d^	6.14 ± 0.48 ^d^	22.63 ± 1.46 ^b^	77.77 ± 3.80 ^c^	11.94 ± 0.70 ^cd^	5.65 ± 0.46 ^ab^
BBC	8.32 ± 0.02 ^a^	8.35 ± 0.26 ^ab^	21.93 ± 0.40 ^b^	117.46 ± 6.10 ^a^	11.31 ± 0.75 ^d^	4.98 ± 0.21 ^bc^
FeBBC	7.82 ± 0.01 ^e^	6.75 ± 0.13 ^cd^	25.43 ± 1.07 ^a^	80.51 ± 7.34 ^c^	12.76 ± 0.42 ^bc^	5.86 ± 0.14 ^a^

Note: AP, available P; AN, alkali-hydro nitrogen; AK, available K; AFe, available Fe; Fed, free Fe oxides; CK: control; BC: *Bacillus clausii*; B: biochar; FeB: Fe-modified biochar; BBC: biochar combined with *Bacillus clausii*; FeBBC: Fe-modified biochar combined with *Bacillus clausii*.

**Table 3 biology-13-00611-t003:** Changes in the richness and diversity of soil microbial communities.

Sample	ASVs	Ace	Chao	Goods Coverage (%)	Shannon	Simpson
CK	3916	3994.39	3937.67	99.59	7.47	0.0012
BC	3690	3734.58	3702.95	99.73	7.59	0.0009
B	3536	3556.61	3538.84	99.87	7.52	0.0009
FeB	3453	3453.00	3453.00	100.00	7.47	0.0011
BBC	3424	3461.45	3431.95	99.78	7.41	0.0012
FeBBC	4077	4113.06	4081.85	99.80	7.61	0.0009

Note: ASVs: Amplicon sequence variants; Ace: Ace index; Chao: Species richness estimator; Goods coverage: Microbial analysis depth; Shannon: The Shannon index; Simpson: The Simpson index. CK: control; BC: *Bacillus clausii*; B: biochar; FeB: Fe-modified biochar; BBC: biochar combined with *Bacillus clausii*; FeBBC: Fe-modified biochar combined with *Bacillus clausii*.

## Data Availability

The datasets generated during and/or analyzed during the current study are available from the corresponding author on reasonable request.

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
