# Peer review of "Enhanced Soil Fertility and Carbon Sequestration in Urban Green Spaces through the Application of Fe-Modified Biochar Combined with Plant Growth-Promoting Bacteria"

_biology, 2024, doi:10.3390/biology13080611_

Round 1
Reviewer 1 Report
Comments and Suggestions for Authors
Major Revision recommendation
Line 19. “Improved structure” is not Inappropriate here.
Line 26. What’s the NCBI accession number of Bacillus clausii ?
Line 29 - Line 30. The introduction of your experimental design is missing before the Results section.
Line 36. “suitable living environemnt for BC” is not logical consistency here.
Line 48. How soil captures CO2? Alternatively, it could be CO2 sedimentation.
Line 58. The correct expression should be that “biochar has a very high specific surface area” instead of “low surface area.”
Line 59. What's the “constrained effect of alkaline biochar” means here?
Line 68 – line 77. The authors should concentrate more on carbon sequestration, instead of “plant growth promotion only”.
Line 74 & Line 76. Citation error.
Line 84. What is the meaning of repair materials?
Line 86. Is there a contradiction with the previous statement about the "low surface area of biochars"?
Line 89- line 90. The authors need to rewrite this part.
Line 96 -line 105. As a reviewer, I still have no idea of the differences between biochar and Fe-modified biochar.
Line 120. NCBI accession number should be provided.
Line 124 – line 131. The authors need to rewrite this part.
Line 138. BC is usually abbreviated as biochar.
Line 139. What is the concentration of PGPB?
Line 240. 1. units are placed in parentheses; 2. a three-line table is required.
Line 256. Error bars are unclear.
Line 323. The characteristics in the figure are unclear.
Line 474. Italicized Latin should not be used for microbial names at the phylum level.
Overall, many language and logic errors have occurred in this manuscript. The authors will require time for thorough revision.
Comments on the Quality of English LanguageModerate editing of English language required to improve this manuscript.
Author Response
Responses to comments of the reviewers 1
Manuscript ID: biology-3145422
Title: Enhanced soil fertility and carbon sequestration in urban green spaces through the application of Fe-modified biochar combined with plant growth promoting bacteria
Authors: Guoyao Niu, Chiquan He, Shaohua Mao, Zongze Chen, Yangyang Ma, Yi Zhu
Dear Zeta Chen,
On behalf of my co-authors, we thank you very much for giving us an opportunity to revise our manuscript, we appreciate editor and reviewers very much for their positive and constructive comments and suggestions on our manuscript entitled “Enhanced soil fertility and carbon sequestration in urban green spaces through the application of Fe-modified biochar combined with plant growth promoting bacteria”. (Ms. Ref. No.: biology-3145422)
We have studied reviewer’s comments carefully and have made revisions which marked in blue in the manuscript. We have tried our best to revise our manuscript according to the comments. Attached please find the revised version, which we would like to submit for your kind consideration. Looking forward to hearing from you.
Yours sincerely,
Chiquan He
August.05,2024
1. Comments of reviewer #1 and author’s response
Comments to the Author – Major Revision recommendation.
Response: We would like to thank the reviewer for commenting on the manuscript carefully and kindly. We have carefully read the suggestions put forward by the reviewers and provided detailed responses to each comment.
Comments:
Q1-1 Line 19. “Improved structure” is not Inappropriate here.
Original in P1 lines 16 ~ 19:
The study results demonstrate that in comparison to the individual application of FeB and BC, the FeBBC treatment significantly relieve soil alkalization, enhances soil alkali-hydro nitrogen content and aggregate stability (particle size > 0.25 mm), thereby contributing to improved soil fertility and structure.
Original in P1 lines 31 ~ 33:
Compared to the individual application of FeB and BC, the FeBBC treatment significantly improved aggregates' stability and positively improved soil fertility and structure.
Revised in P1 lines 16 ~ 19:
The study results demonstrate that in comparison to the individual application of FeB and BC, the FeBBC treatment significantly relieve soil alkalization, enhances soil alkali-hydro nitrogen content and aggregate stability (particle size > 0.25 mm), thereby contributing to improved soil fertility and ecological function.
Revised in P1 lines 44 ~ 46:
Compared to the individual application of FeB and BC, the FeBBC treatment significantly improved aggregates' stability and positively improved soil fertility and ecological function.
Q1-2 Line 26. What’s the NCBI accession number of Bacillus clausii ?
Response: In this study, The Bacillus clausii CICC 21104 was provided by China Center of Industrial Culture Collection (http://www.china-cicc.org/cicc/detail2/?sid=3305), Beijing, China. Bacillus clausii CICC 21104 is native to China and is now a mature commercial strain. However, due to the lack of regulations on uploading NCBI database platform data on the China Center of Industrial Culture Collection platform. Therefore, the platform can only provide proof of strain origin and does not have an NCBI accession number.
Q1-3 Line 29 - Line 30. The introduction of your experimental design is missing before the Results section.
Response: We greatly appreciate the reviewer's suggestions and have made revisions based on them.
Original in P1 lines 28 ~ 30:
Additionally, the effects on C-cycling-related enzyme activity and bacterial community were also explored. Compared with other treatments, the Fe-modified biochar combined with Bacillus clausii (FeBBC) treatment significantly decreased soil pH, alleviated soil alkalization and increased the alkali-hydro nitrogen content in the soil.
Revised in P1 lines 39 ~ 44:
Additionally, the effects on C-cycling-related enzyme activity and bacterial community were also explored. Six treatments include: no biochar or Bacillus clausii suspension added (CK), only Bacillus clausii suspension (BC), only biochar (B), only Fe-modified biochar (FeB), biochar combined with Bacillus clausii (BBC), and Fe-modified biochar combined with Bacillus clausii (FeBBC). Compared with other treatments, the Fe-modified biochar combined with Bacillus clausii (FeBBC) treatment significantly decreased soil pH, alleviated soil alkalization and increased the alkali-hydro nitrogen content in the soil.
Q1-4 Line 36. “suitable living environemnt for BC” is not logical consistency here.
Original in P1 lines 35 ~ 38:
In FeBBC treatment, FeB could provide a suitable living environment for BC or microorganisms in soil, increase the abundance and diversity of bacterial communities, and promote the uniform distribution of carbon-related bacteria in the soil.
Revised in P2 lines 49 ~ 52:
In FeBBC treatment, FeB can improve soil physicochemical properties and provide bacterial attachment sites, increase the abundance and diversity of bacterial communities, and promote the uniform distribution of carbon related bacteria in the soil.
Q1-5 Line 48. How soil captures CO2? Alternatively, it could be CO2 sedimentation.
Original in P2 lines 47 ~ 49:
Soil in urban green spaces can increase its nutrient content and enhance carbon storage capacity by capturing carbon dioxide from the atmosphere or decomposing plant waste materials.
Revised in P2 lines 61 ~ 63:
The soil in urban green spaces can increase its nutrient content and enhance its carbon storage capacity through atmospheric carbon dioxide precipitation or decomposition of plant waste.
Q1-6 Line 58. The correct expression should be that “biochar has a very high specific surface area” instead of “low surface area.”
Response: We greatly appreciate your suggestions and have made revisions to the manuscript based on them.
Revised in P2 lines 70 ~ 79:
Biochar is a solid carbonaceous material produced by converting carbon rich biomass through thermochemical methods under anaerobic or hypoxic conditions. Previous studies have demonstrated that adding biochar to soil effectively improves soil fertility and increases carbon sequestration [6, 7]. Recent research has focused on the functional modification of biochar to enhance its potential applications in carbon sequestration, soil fertility improvement, and environmental pollution remediation [8, 9].
Q1-7 Line 59. What's the “constrained effect of alkaline biochar” means here?
Response: We greatly appreciate your suggestions and have made revisions to the manuscript based on them.
Revised in P2 lines 70 ~ 79:
Biochar is a solid carbonaceous material produced by converting carbon rich biomass through thermochemical methods under anaerobic or hypoxic conditions. Previous studies have demonstrated that adding biochar to soil effectively improves soil fertility and increases carbon sequestration [6, 7]. Recent research has focused on the functional modification of biochar to enhance its potential applications in carbon sequestration, soil fertility improvement, and environmental pollution remediation [8, 9].
Q1-8 Line 68 – line 77. The authors should concentrate more on carbon sequestration, instead of “plant growth promotion only”.
Original in P2 lines 68 ~ 77:
Plant growth promoting bacteria (PGPB) can enhance plant growth and contribute to soil nutrient cycling through diverse mechanisms, including phosphorus solubilization, nitrogen fixation, and siderophore production [14]. Bacillus sp. is the bacterium that is most frequently utilized for promoting plant growth. Some studies have demonstrated that the inoculation of Bacillus sp. can effectively increase soil nutrient and carbon content [15]. Bacillus clausii is a microorganism that has been recognized for its ability to promote plant growth. Xiangyong, et al. [16] confirmed the production of extracellular polysaccharides by Bacillus clausii isolated from saline alkali soil in Xinjiang. The Bacillus clausii B8 strain isolated from soil by Oulebsir-Mohandkaci et al. [17] can produce siderophore, HCN, and IAA, leading to a significant improvement in rapeseed seed germination rate.
Revised in P2 lines 84 ~ 96:
Plant Growth Promoting Bacteria (PGPB) can promote plant growth through direct and indirect effects, thereby increasing soil carbon sequestration [14]. PGPB can directly promote plant growth through the synthesis of growth hormones (IAA) or through the effects of phosphorus solubilization, nitrogen fixation, and potassium solubilization [15]. In addition, PGPB can indirectly promote plant growth by improving soil structure and inhibiting the growth of pathogens [16]. Bacillus sp. is the bacterium that is most frequently utilized for promoting plant growth. Some studies have demonstrated that the inoculation of Bacillus sp. can effectively increase soil nutrient and carbon sequestration [17]. Bacillus clausii is a microorganism that has been recognized for its ability to promote plant growth. Li, et al. [18] confirmed the production of extracellular polysaccharides by Bacillus clausii isolated from saline alkali soil in Xinjiang. The Bacillus clausii B8 strain isolated from soil by Oulebsir-Mohandkaci et al. [19] can produce siderophore, HCN, and IAA, leading to a significant improvement in rapeseed seed germination rate.
Q1-9 Line 74 & Line 76. Citation error.
Original in P2 lines 74 ~ 78:
Xiangyong, et al. [16] confirmed the production of extracellular polysaccharides by Bacillus clausii isolated from saline alkali soil in Xinjiang. The Bacillus clausii B8 strain isolated from soil by Oulebsir-Mohandkaci et al. [17] can produce siderophore, HCN, and IAA, leading to a significant improvement in rapeseed seed germination rate.
Revised in P2 lines 93 ~ 96:
Li, et al. [18] confirmed the production of extracellular polysaccharides by Bacillus clausii isolated from saline alkali soil in Xinjiang. The Bacillus clausii B8 strain isolated from soil by Oulebsir-Mohandkaci et al. [19] can produce siderophore, HCN, and IAA, leading to a significant improvement in rapeseed seed germination rate.
Q1-10 Line 84. What is the meaning of repair materials?
Response: In this study, repair materials refer to materials used to restore soil ecological functions, which play an important role in improving environmental quality and protecting the ecological environment. For example, biochar, organic fertilizers, microbial agents, and so on.
Original in P2 lines 83 ~ 84:
However, comparing the effects between biotic and abiotic repair materials has not been fully studied.
Revised in P3 lines 101 ~ 103:
However, the comparison of the effects between biotic and abiotic soil ecological remediation materials has not been fully studied.
Q1-11 Line 86. Is there a contradiction with the previous statement about the "low surface area of biochars"?
Response: We fully understand the concerns of the reviewers and fully recognize the issues raised in the wording. Therefore, we have made corrections to this in the manuscript.
Q1-12 Line 89- line 90. The authors need to rewrite this part.
Original in P2 lines 89 ~ 90:
For example, Jabborova et al. [22] indicated that the higher content of SOC and nutrients in soil in the biochar and B. japonicum USDA 110 treatment.
Revised in P3 lines 107 ~ 109:
For example, Jabborova et al. [24] indicated that co-inoculation of biochar and rhizobacteria significantly increased soil SOC and nutrient content.
Q1-13 Line 96 -line 105. As a reviewer, I still have no idea of the differences between biochar and Fe-modified biochar.
Response: We greatly appreciate the reviewer's suggestions and have made corrections in the manuscript. The difference between biochar and iron modified biochar is shown in line 67-80 of the manuscript. In addition, we have determined the differences in properties between biochar and Fe-modified biochar, as shown in the supplementary materials.
Q1-14 Line 120. NCBI accession number should be provided.
Response: In this study, The Bacillus clausii CICC 21104 was provided by China Center of Industrial Culture Collection (http://www.china-cicc.org/cicc/detail2/?sid=3305), Beijing, China. Bacillus clausii CICC 21104 is native to China and is now a mature commercial strain. However, due to the lack of regulations on uploading NCBI database platform data on the China Center of Industrial Culture Collection platform. Therefore, the platform can only provide proof of strain origin and does not have an NCBI accession number.
Q1-15 Line 124 – line 131. The authors need to rewrite this part.
Original in P3 lines 124 ~ 131:
CM0002 nutrient broth culture medium was the medium used for culture of Bacillus clausii. Culture medium: 1.25 g peptone, 0.75 g beef extract, 1.25 g NaCl, 250 mL distilled water, pH 7.0. Inoculate Bacillus clausii into sterile medium to cultivate and expand at 37℃ for 48 h. Centrifugation of the bacterial culture in a sterile centrifuge tube at 4℃ and 6,000 rpm for 10 min took place and the supernatant was discarded. Disperse with sterile water, centrifuge, and repeat this process twice. Finally, dilute the residue with sterile water and adjust the concentration of Bacillus clausii suspension to 1 × 108 CFU mL-1 by using OD600 = 1.
Revised in P3 lines 143 ~ 146:
Nutrient broth (CM0002, China Center of Industrial Culture Collection, China.) culture medium was the medium used for culture of Bacillus clausii. Bacillus clausii was inoculated into sterile medium to cultivate and expand at 37℃ for 48 h. We regulated the suspension of Bacillus clausii to 1 × 108 CFU mL-1, and then used it as a standard inoculum.
Q1-16 Line 138. BC is usually abbreviated as biochar.
Response: We really appreciate the questions raised by the reviewers. We have provided a detailed explanation as follows. Before the experiment began, we conducted extensive literature research, as shown in Table. We found that in research related to biochar, biochar is abbreviated as B, and modified biochar is abbreviated as M-B. Therefore, we also abbreviated biochar as B and have been using it ever since. To distinguish it from biochar (B), we will abbreviate Bacillus clausii as BC and Fe-modified biochar as FeB..
Table
|
Title |
Abbreviation |
Reference |
|
Effects of biochar and organic substrates on biodegradation of polycyclic aromatic hydrocarbons and microbial community structure in PAHs-contaminated soils |
biochar (B) and compost (CP), mushroom residue (M) and corn straw (Y) |
[1] |
|
Amelioration of calcareous sandy soil productivity via incorporation between biochar and some organic manures |
biochar (B) with farmyard manure (FYM) and poultry manure (PM) |
[2] |
|
Water De-Chlorination by Non-Modified and Modified Biochar Derived from Date Palm |
Non-modified biochar (NM-B) and modified biochar (M-B) |
[3] |
|
Nitrogen combined with biochar changed the feedback mechanism between soil nitrification and Cd availability in an acidic soil |
Biochar (B) and N fertilizers with biochar (N-B) |
[4] |
Reference:
- Bao, H.; Wang, J.; Zhang, H.; Li, J.; Li, H.; Wu, F., Effects of biochar and organic substrates on biodegradation of polycyclic aromatic hydrocarbons and microbial community structure in PAHs-contaminated soils.J Hazard Mater2020, 385, 121595. http://dx.doi.org/10.1016/j.jhazmat.2019.121595
- Amin, A. E.-E. A. Z., Amelioration of calcareous sandy soil productivity via incorporation between biochar and some organic manures. Arabian J. Geosci.2018,11, (23). http://dx.doi.org/10.1007/s12517-018-4133-y
- Alfaiz, S. K.; Yaseen, D. A.; Alawadi, W. A., Water De-Chlorination by Non-Modified and Modified Biochar Derived from Date Palm. Journal of Ecological Engineering2023, 24,(12), 377-387. http://dx.doi.org/10.12911/22998993/173490
- Zhao, H.; Yu, L.; Yu, M.; Afzal, M.; Dai, Z.; Brookes, P.; Xu, J., Nitrogen combined with biochar changed the feedback mechanism between soil nitrification and Cd availability in an acidic soil.J. Hazard. Mater.2020, 390. http://dx.doi.org/10.1016/j.jhazmat.2019.121631
Q1-17 Line 139. What is the concentration of PGPB?
Response: The concentration of Bacillus clausii suspension is 1 × 108 CFU mL-1. For the Bacillus clausii (BC) treatment, we diluted 10 mL of Bacillus clausii suspension with sterile water to 100 mL, and then fully mixed with the soil in the pot.
Q1-18 Line 240. 1. units are placed in parentheses; 2. a three-line table is required.
Response: We greatly appreciate the reviewer's suggestions and have made corrections in the manuscript based on their suggestions.
Q1-19 Line 256. Error bars are unclear.
Response: We greatly appreciate the reviewer's suggestions and have bolded the error bars in the images based on their suggestions.
Q1-20 Line 323. The characteristics in the figure are unclear.
Response: We greatly appreciate the reviewers' suggestions and have made corrections to the manuscript based on their suggestions, using clearer images. However, the PDF output from the submission system may reduce the clarity of the images, so we have uploaded a separate image file.
Q1-21 Line 474. Italicized Latin should not be used for microbial names at the phylum level.
Response: We greatly appreciate the reviewer's suggestions and have made corrections in the manuscript based on their suggestions.

Reviewer 2 Report
Comments and Suggestions for Authors
Submission ID: biology-3145422
Title of the manuscript: "Enhanced soil fertility and carbon sequestration in urban green spaces through the application of Fe-modified biochar combined with plant growth promoting bacteria".
This study evaluated the impact of plant growth promoting bacteria (Bacillus clausii) and Fe-modified biochar on soil fertility and carbon sequestration. The subject is very interesting, and the methodology used is adequate for the objectives of the study. The results are of interest and support the conclusions. The manuscript is worth publishing in “Biology”. However, there are still some issues that need to be addressed, I suggest major revision.
Specific comments
L41: Please arrange the keywords in alphabetical order.
L54: What did the authors mean by “degradation of soil fertility”? Not clear.
L56: Give some information about biochar such as its definition-composition to the readers.
L58-70: Not sufficient. The role of PGPB in enhancing plant growth and productivity including many more mechanisms. Please extend these mechanisms.
Sometimes authors use PGPB and in other part use PGPR, please use one form in the whole manuscript.
L121: Is this the accession number of the bacterial strain in the GenBank?
L123: Bacterial inoculum preparation.
L124: Nutrient broth (CM0002, OXOID, UK).
L125: Delete the composition of the medium. Not necessary.
L126: Bacillus clausii was inoculated…………..
L127: Replace “rpm” with “xg”. RCF = (RPM)2 × 1.118 × 10-5 × r
L163-185: Please briefly provide references to these measurements.
L196: P should be italic in the whole manuscript.
L194-204: Which program was used by the authors for SA.
Figure 4a: Please enhance the quality.
L220: I could not find the supplementary data in this submission.
L218-220: The presentation of the key findings of experimental results should be improved.
I suggest that the authors could include a list of abbreviations as the manuscript is full of abbreviations.
L351: I appreciate the well written discussion, however, please try to make the titles of the discussion represent the content. I suggest “FeB combined with BC improve soil physicochemical properties and fertility” and throughout the discussion.
L511-514: Move this paragraph to the conclusion section.
L516-531: Instead of repeating the results here, please focus on work novelty, future prospectives, how these results could be important for sustainable agriculture.
Regards.
Comments on the Quality of English LanguageMinor editing of English language required
Author Response
Responses to comments of the reviewers 2
Manuscript ID: biology-3145422
Title: Enhanced soil fertility and carbon sequestration in urban green spaces through the application of Fe-modified biochar combined with plant growth promoting bacteria
Authors: Guoyao Niu, Chiquan He, Shaohua Mao, Zongze Chen, Yangyang Ma, Yi Zhu
Dear Zeta Chen,
On behalf of my co-authors, we thank you very much for giving us an opportunity to revise our manuscript, we appreciate editor and reviewers very much for their positive and constructive comments and suggestions on our manuscript entitled “Enhanced soil fertility and carbon sequestration in urban green spaces through the application of Fe-modified biochar combined with plant growth promoting bacteria”. (Ms. Ref. No.: biology-3145422)
We have studied reviewer’s comments carefully and have made revisions which marked in blue in the manuscript. We have tried our best to revise our manuscript according to the comments. Attached please find the revised version, which we would like to submit for your kind consideration. Looking forward to hearing from you.
Yours sincerely,
Chiquan He
August.05,2024
1. Comments of reviewer #2 and author’s response
This study evaluated the impact of plant growth promoting bacteria (Bacillus clausii) and Fe-modified biochar on soil fertility and carbon sequestration. The subject is very interesting, and the methodology used is adequate for the objectives of the study. The results are of interest and support the conclusions. The manuscript is worth publishing in “Biology”. However, there are still some issues that need to be addressed, I suggest major revision.
Response: We would like to thank the reviewer for commenting on the manuscript carefully and kindly. We have carefully read the suggestions put forward by the reviewers and provided detailed responses to each comment.
Comments:
L41: Please arrange the keywords in alphabetical order.
Original in P1 lines 41:
Keywords: soil restoration; carbon sequestration; enzyme activity; Fe-modified biochar; soil fertility
Revised in P2 lines 55:
Keywords: carbon sequestration; enzyme activity; Fe-modified biochar; soil fertility; soil restoration
L54: What did the authors mean by “degradation of soil fertility”? Not clear.
Response: The degradation of soil fertility in urban green spaces refers to the process in which the productivity, environmental regulation potential, and sustainable development capacity of soil in urban environments are reduced or even completely lost due to natural factors and human activities. This usually manifests as a decrease in soil physical, chemical, and biological quality, such as soil structure destruction, nutrient loss, soil compaction, acidification, salinization, and other phenomena. The degradation of soil fertility directly affects the growth of plants, which in turn affects the ecological benefits and landscape functions of urban green spaces.
L56: Give some information about biochar such as its definition-composition to the readers.
Response: We greatly appreciate the reviewer's suggestions and have supplemented the definition of biochar as follows.
Revised in P2 lines 70 ~ 73:
Biochar is a solid carbonaceous material produced by converting carbon rich biomass through thermochemical methods under anaerobic or hypoxic conditions. Previous studies have demonstrated that adding biochar to soil effectively improves soil fertility and increases carbon sequestration [6, 7].
L68-70: Not sufficient. The role of PGPB in enhancing plant growth and productivity including many more mechanisms. Please extend these mechanisms. Sometimes authors use PGPB and in other part use PGPR, please use one form in the whole manuscript.
Original in P2 lines 68 ~ 70:
Plant growth promoting bacteria (PGPB) can enhance plant growth and contribute to soil nutrient cycling through various mechanisms, including phosphorus solubilization, nitrogen fixation, and siderophore production [14].
Revised in P2 lines 84 ~ 89:
Plant Growth Promoting Bacteria (PGPB) can promote plant growth through direct and indirect effects, thereby increasing soil carbon sequestration [14]. PGPB can directly promote plant growth through the synthesis of growth hormones (IAA) or through the effects of phosphorus solubilization, nitrogen fixation, and potassium solubilization [15]. In addition, PGPB can indirectly promote plant growth by improving soil structure and inhibiting the growth of pathogens [16].
L121: Is this the accession number of the bacterial strain in the GenBank?
Response: In this study, The Bacillus clausii CICC 21104 was provided by China Center of Industrial Culture Collection (http://www.china-cicc.org/cicc/detail2/?sid=3305), Beijing, China. Bacillus clausii CICC 21104 is native to China and is now a mature commercial strain. However, due to the lack of regulations on uploading NCBI database platform data on the China Center of Industrial Culture Collection platform. Therefore, the platform can only provide proof of strain origin and does not have an NCBI accession number.
L123: Bacterial inoculum preparation.
Original in P3 lines 123:
Culture preparation
Revised in P3 lines 142:
Bacterial inoculum preparation
L124: Nutrient broth (CM0002, OXOID, UK).
Original in P3 lines 124:
CM0002 nutrient broth culture medium was the medium used for culture of Bacillus clausii.
Revised in P3 lines 143 ~ 144:
Nutrient broth (CM0002, China Center of Industrial Culture Collection, China.) culture medium was the medium used for culture of Bacillus clausii.
L125: Delete the composition of the medium. Not necessary.
Response: We greatly appreciate the reviewer's suggestions and have made corrections based on them.
L126: Bacillus clausii was inoculated…………..
Original in P3 lines 126 ~ 127:
Inoculate Bacillus clausii into sterile medium to cultivate and expand at 37℃ for 48 h.
Revised in P3 lines 144 ~ 145:
Bacillus clausii was inoculated into sterile medium to cultivate and expand at 37℃ for 48 h.
L127: Replace “rpm” with “xg”. RCF = (RPM)2 × 1.118 × 10-5 × r
Response: We greatly appreciate the reviewer's suggestions and have made corrections based on them. Using a MGL-16MA centrifuge, centrifuge the bacterial culture at 4 ℃ and approximately 3903 xg for 10 min, then disperse it in sterile water, centrifuge, and repeat the process twice.
L163-185: Please briefly provide references to these measurements.
Response: We greatly appreciate the reviewer's suggestions and have made revisions based on them. The measurement method for the characteristics of Bacillus clausii can be found in lines 144-180 of the supplementary materials.
L196: P should be italic in the whole manuscript.
Response: We greatly appreciate your suggestion and have made corrections in the manuscript. Please refer to the manuscript for details.
L194-204: Which program was used by the authors for SA.
Response: We greatly appreciate the reviewer's suggestions and have made revisions based on them. We have resubmitted the supplementary materials. Detailed bacterial community analysis methods can be found in lines 102 ~ 119 of the supplementary materials.
Analytic method for bacterial community analysis:
The quality and concentration of the extracted DNA were assessed by performing 1% agarose gel electrophoresis. The concentration and purity of DNA can be determined by using NanoDrop2000 (Thermo Scientific, USA). All PCR reactions were carried out in 20 μL volumes containing 4 μL 5 × Fast Pfu buffer, 2 μL 2.5 mM dNTPs, 0.8 μL forward and reverse primers (5 μM), 0.4 μL Fast Pfu Polymerase, 0.2 μL BSA and about 10 ng of template DNA, finally supplemented with ddH2O to 20 μL. Thermal cycling consisted of an initial denaturation at 95 °C for 3 min followed by 27 cycles of denaturation at 95 °C for 30 s, annealing at 55 °C for 30 s, and elongation at 72 °C for 45 s, and a final step of 72 °C for 10 min, 10 °C until halted by user. Recover PCR products using a 2% agarose gel and purify them using a PCR Clean-Up Kit (Yuhua, Shanghai, China). The purified PCR products were subjected to paired-end sequencing on an Illumina PE300 platform (Illumina, San Diego, USA) according to the standard protocols by Majorbio Bio-Pharm Technology Co. Ltd. (Shanghai, China). The obtained sequencing reads were demultiplexed were quality filtered with fastp (v0.19.6) and then merged with flash (v1.2.11). Then the high-quality sequences were de-noised using the QIIME2 pipeline with recommended parameters, which obtains single nucleotide resolution based on error profiles within samples. The Majorbio Cloud platform (https://cloud.majorbio.com) was utilized for the bioinformatic analysis of soil microbiota. Based on the ASVs information, alpha diversity indices including observed Chao1 richness, Shannon index and Good’s coverage were calculated with Mothur v1.30.1.
Figure 4a: Please enhance the quality.
Response: We greatly appreciate the reviewer's suggestions and have made improvements to the quality of Figure 4a based on their suggestions.
L220: I could not find the supplementary data in this submission.
Response: We greatly appreciate the reviewer's suggestions and have made revisions based on them. We have resubmitted the supplementary materials.
L218-220: The presentation of the key findings of experimental results should be improved.
I suggest that the authors could include a list of abbreviations as the manuscript is full of abbreviations.
Response: We greatly appreciate the reviewer's suggestions and have made revisions accordingly. Please refer to the manuscript for details. Please refer to lines 24-34 of the manuscript for the abbreviation list.
Original in P5 lines 218 ~ 220:
The results showed that Bacillus clausii could produce indole-3-acetic acid (IAA), produce ammonia, solubilize phosphorus, fix nitrogen, produce siderophores, and produce carbonic anhydrase activity (Table S3).
Revised in P6 lines 233 ~ 235:
The results showed that Bacillus clausii has various plant growth promoting characteristics, such as indole-3-acetic acid (IAA) production, phosphorus solubilization, nitrogen fixation, siderophores production, and carbonic anhydrase production (Table S3).
L351: I appreciate the well written discussion, however, please try to make the titles of the discussion represent the content. I suggest “FeB combined with BC improve soil physicochemical properties and fertility” and throughout the discussion.
Response: We greatly appreciate your suggestion and have made revisions based on it.
L511-514: Move this paragraph to the conclusion section.
Response: We greatly appreciate your suggestion and have made revisions based on it. Please refer to the manuscript for details.
L516-531: Instead of repeating the results here, please focus on work novelty, future prospectives, how these results could be important for sustainable agriculture.
Original in P14 lines 516 ~ 531:
In conclusion, this study highlights that the combination of FeB and BC had a significant positive impact on enhancing the soil fertility of urban green spaces and increasing soil carbon sequestration. This combination effectively mitigated soil alkalization and enhanced the availability of soil nutrients, particularly alkali-hydro nitrogen. Additionally, it promoted the formation of aggregates with a particle size greater than 0.25 mm. Furthermore, the combination of FeB and BC exhibited a remarkable increase in soil organic carbon content, as well as an enhancement in the richness and diversity of bacterial communities in the soil. Moreover, this combined approach also leads to improvements in the composition of C-cycling-related bacterial communities, thereby positively influencing soil stability and promoting a balanced ecosystem. Overall, our study findings clearly demonstrated that the combined application of FeB and BC had a greater potential in enhancing the soil fertility of urban green spaces and increasing soil carbon sequestration compared to their individual applications. These results provide valuable insights into the effective utilization of Fe-modified biochar combined with Bacillus clausii in sustainable soil management practices and carbon sequestration efforts.
Revised in P14 lines 530 ~ 542:
In conclusion, this study innovatively combined Fe-modified biochar (FeB) with Bacillus clausii (BC) to restore the ecological function of urban green soil. The results showed that the combination of FeB and BC effectively alleviates soil alkalization, improves soil fertility, and increases soil carbon sequestration. Additionally, the combination of FeB and BC significantly increased the activity of invertase and the richness and diversity of bacterial communities in the soil, promoting the uniform distribution of carbon cycle related bacteria in the soil. The results clearly indicate that the combined application of FeB and BC has greater potential in improving soil fertility and increasing soil carbon sequestration in urban green spaces compared to their individual applications. These results provide a reference solution for the future restoration of soil ecological functions, and provide a solid theoretical basis for restoring the ecological functions of urban green space soil and achieving sustainable management of urban or agricultural soil.

Round 2
Reviewer 1 Report
Comments and Suggestions for Authors
The authors have significantly improved the manuscript and answered my questions. The manuscript can be accepted in its current form.
Comments on the Quality of English LanguageVery minor editing of English language required.
Reviewer 2 Report
Comments and Suggestions for Authors
This is the second time I have evaluated this manuscript. The authors addressed all my comments, and the manuscript has been noticeably improved. Many thanks for their contribution.
Note: L376,424,468: Delete "Effect" and replace with "The application".
Comments on the Quality of English LanguageThe English language is understandable and correct. Only minor editorial and stylistic corrections are required.